# Theory predicts UV/vis-to-IR photonic down conversion mediated by excited state vibrational polaritons

Connor K. Terry Weatherly[1] ✉, Justin Provazza[1] ✉, Emily A. Weiss [1] ✉ & Roel Tempelaar [1] ✉

This work proposes a photophysical phenomenon whereby ultraviolet/visible (UV/vis) excitation of a molecule involving a Franck-Condon (FC) active vibration yields infrared (IR) emission by strong coupling to an optical cavity. The resulting UV/vis-to-IR photonic down conversion process is mediated by vibrational polaritons in the electronic excited state potential. It is shown that the formation of excited state vibrational polaritons (ESVP) via UV/vis excitation only involve vibrational modes with both a non-zero FC activity and IR activity in the excited state. Density functional theory calculations are used to identify 1-Pyreneacetic acid as a molecule with this property and the dynamics of ESVP are modeled. Overall, this work introduces an avenue of polariton chemistry where excited state dynamics are influenced by the formation of vibrational polaritons. Along with this, the UV/vis-to-IR photonic down conversion is potentially useful in both sensing excited state vibrations and quantum transduction schemes.

Most light–matter interactions involved in spectroscopy are in the weak coupling regime such that the intramolecular Coulombic fields are significantly stronger than the interacting radiation fields[1]. As such, perturbative theories accurately describe many spectroscopic phenomena[2]. These theories break down in the "strong coupling" regime where light and matter states hybridize, resulting in energy splittings that can be spectroscopically observed[3]. The quanta of such hybrid light–matter states are called "polaritons", the formation of which is commonly realized by the use of high-finesse optical cavities with quantized radiation modes that can be coupled to dipole-carrying transitions within states of matter. Polaritons formed between infrared (IR)-active molecular or condensed phase vibrations and an optical cavity are, naturally, called "vibrational polaritons".

In recent experiments, it has been demonstrated that product selectivity and rates of ground-state reactions may be modulated by the formation of vibrational polaritons through coupling an optical cavity to the vibrational coordinate on which a reaction proceeds[4–10]. Although there has been significant experimental and theoretical work towards understanding such ground-state vibrational polaritons and

their effect on reactions along with other physiochemical phenomena[3,11–18], analogies on the electronic excited-state potential have remained underexplored[19,20]. Here, we propose a photonic downconversion process that results from the formation of excited-state vibrational polaritons (ESVP) via UV/vis electronic excitation of a molecule containing a vibrational mode that is coupled to an IR cavity. Under vibronic coupling, the UV/vis excitation of the molecule induces a wavepacket consisting of ESVP, while Rabi oscillations occur between occupied vibrational and photonic states as time evolves. These photons in turn may leak from the cavity, leading to IR light emission[21].

The UV/vis-to-IR photonic downconversion process proposed in this work has potential applications in quantum information science, where the vibrational wave-packet, formed by a UV/vis excitation of the cavity-embedded molecule, can be directly transduced into a photonic wavepacket through coupling to the optical cavity. Hence, molecules with different vibronic couplings and excited-state potential energy surfaces produce different coherences within the photonic wavepacket emitted via the ESVP photonic downconversion process. This could allow for the manipulation of coherent light by molecular

[1]Department of Chemistry, Northwestern University, Evanston, IL 60208-3113, USA. ✉e-mail: connorterryweatherly2025@u.northwestern.edu; jprov410@gmail.com; e-weiss@northwestern.edu; roel.tempelaar@northwestern.edu

design. Furthermore, the UV/vis-to-IR photonic downconversion process implies that an optical cavity can be used to enhance the emission of excited-state vibrations, analogous to the Purcell effect in the weak light–matter coupling regime[22,23]. The effective detection of excited-state vibrations through this process would be a powerful measurement tool for identifying vibrational modes induced by electronic transitions—information that has been difficult to obtain experimentally[24], and that often requires a determination through electronic structure calculations and molecular dynamics simulations[25]. Detection of photonic emission from vibrations is typically impossible due to their non-radiative relaxation being on the order of picoseconds[26,27], which outcompetes any non-enhanced emission that occurs on a >1 μs time scale[28,29]. It was experimentally indicated by Raschke and coworkers that ground-state vibrational emission could be enhanced by coupling molecules to an optical resonator, yielding a 50% decrease in the vibrational dephasing lifetime for poly(methylmethacrylate)[29], although photonic emission from vibrations was not actually observed.

Another potential application of ESVP is in the modulation of photophysical processes that occur via transitions between excited electronic states, and which can be mediated by vibrational coherences. For example, vibrational coherences have been shown to mediate singlet fission in certain molecular systems[30–35], promote charge and energy transfer in biological light-harvesting systems[36–39], as well as drive the photochemical reaction of vision[40–42]. The hybridization of cavity photons with relevant vibrational modes could modulate such coherences as well as change excited-state potential energy surfaces (PES), altering reaction pathways and rates similarly to what has been shown for ground-state reactions.

In this work, we utilize the truncated Wigner approximation (TWA)[43] to theoretically predict the time evolution of quantum mechanical observables in a combined cavity–molecule system, either without dissipation or when coupled to dissipative harmonic baths. The TWA is exact for systems governed by linear or quadratic potentials, i.e., systems consisting of harmonic oscillators. Furthermore, the TWA benefits from linear scaling of complexity with increasing number of degrees of freedom (DOF), as opposed to the exponential increase in complexity of the Hilbert space of the system. Therefore, the TWA enables exact modeling of the evolution of polariton wavepackets coupled to baths of harmonic oscillators without requiring a truncated Hilbert space and without tracing out the bath DOF, as is done in standard quantum master equation techniques[44,45]. We are therefore able to concurrently and explicitly monitor the population transfer into both a radiative and nonradiative bath, giving insights into how different interaction parameters affect photonic emission from ESVP in the presence of nonradiative vibrational relaxation.

## Results

### Photonic downconversion mechanism

Figure 1 shows a schematic of the ESVP-mediated photonic downconversion process. A molecule is positioned inside an optical cavity, initially in its ground electronic state. A mode of the optical cavity is assumed to be near resonant with a spectrally isolated vibrational mode of the molecule. We treat the cavity as transparent to high optical frequencies allowing a UV/vis pulse to promote the molecule into an electronically excited state. We also assume the FC approximation where vibronic coupling is characterized by a linear shift in the excited-state nuclear PES, as shown in Fig. 1, with $Q_0^{(g)}$ and $Q_0^{(e)}$ being the ground and excited-state equilibrium nuclear configurations of the vibrational mode of interest, respectively. Assuming a nonzero Huang–Rhys (HR) factor (vibronic coupling constant), a vibrational wavepacket will be created upon electronic excitation of the molecule. In the absence of the cavity, this wavepacket will have an expected occupation number, $\langle \hat{N}_{vib} \rangle$, equal to the HR factor ($S$)[46,47], and its energy expectation value equals $S\hbar\omega_v$, which is called the reorganization energy, where $\omega_v$ is the vibrational frequency. When the cavity is present and the excited-state vibrational transition dipole moment ($\boldsymbol{\mu}_{IR}^{(e)}$) is nonzero for the cavity resonant vibrational mode, the vibrational wavepacket will comprise a superposition of polariton eigenstates and Rabi oscillations will occur between vibrations and cavity photons as time evolves. Furthermore, given a finite quality factor ($\mathcal{Q}$) of the cavity, which is a measure of its radiative energy loss[48], the cavity photons involved in such ESVP will leak out of the cavity resulting in IR emission that can be detected and/or harnessed[21]. Altogether, these steps constitute a photonic downconversion process where a single incident UV/vis photon is converted into one or more IR photons.

In this work, we only consider the dissipation of cavity photons through the cavity mirrors so that $\mathcal{Q}$ is dependent only on the amount of detectable radiative emission. In general, other dissipation pathways should also be accounted for such as absorption by the cavity mirrors and bound mode photoluminescence[21], which would also decrease $\mathcal{Q}$. Therefore, the results presented herein give an idealized estimate of the amount of photonic downconversion that can occur within a lossy cavity.

### Theoretical model

We use the Pauli–Fierz Hamiltonian as a basis for constructing a model of the ESVP-mediated photonic downconversion process[49–51]. Accordingly, the cavity–molecule system is described as

$$\hat{H}_{system} = \hat{H}_{mol} + \frac{\hat{p}_c^2}{2} + \frac{\omega_c^2}{2}(\hat{q}_c + \mathbf{A} \cdot \hat{\boldsymbol{\mu}})^2, \quad (1)$$

where $\hat{p}_c$ and $\hat{q}_c$ are the cavity mode momentum and position operators, respectively, and $\omega_c$ is the frequency of the cavity mode. Moreover, $\mathbf{A} = \sqrt{2/\hbar\omega_c}A_0\boldsymbol{\epsilon}_c$, where $A_0 = \sqrt{\hbar/2\omega_c\varepsilon V}$ is the amplitude of the cavity mode vector potential, with $\varepsilon$ being the effective permittivity of the cavity and $V$ being the cavity mode volume, and $\boldsymbol{\epsilon}_c$ is the cavity mode polarization unit vector. Lastly, $\hat{\boldsymbol{\mu}}$ is the molecular dipole operator and $\hat{H}_{mol}$ describes the molecule. We express $\hat{H}_{system}$ in the molecular diabatic basis at the ground-state equilibrium, that is, the basis of electronic eigenstates, $|\alpha\rangle$, that satisfy

$$\hat{H}_{mol}^{el}\left(Q_0^{(g)}\right)|\alpha\rangle = E_\alpha\left(Q_0^{(g)}\right)|\alpha\rangle. \quad (2)$$

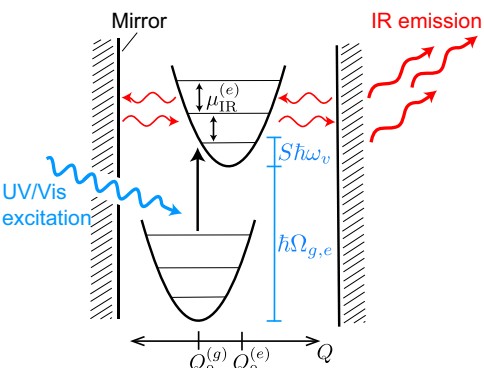

**Fig. 1 | Schematic of the ultraviolet/visible-to-infrared (UV/vis-to-IR) photonic downconversion process mediated by excited-state vibrational polaritons (ESVP).** An impulsive excitation by UV/vis light (blue arrow) interacts with a molecule in it's electronic ground state, promoting the molecule to an electronic excited state through an electronic transition with frequency $\Omega_{e,g}$. Due to vibronic interactions, captured by a nonzero Huang–Rhys factor ($S$), the excitation creates ESVP with energy $S\hbar\omega_v$ along the cavity-coupled vibration with frequency $\omega_v$. The ESVP then leak IR radiation from the cavity (red arrows). Here $\hbar$ is Planck's reduced constant and $Q_g^{(g)}$ ($Q_g^{(e)}$) denotes the ground (excited) state nuclear equilibrium position.

Here, $\hat{H}_{mol}^{el}(Q_0^{(g)})$ is the component of $\hat{H}_{mol}$ containing a dependence on the electronic DOF, parameterized by $Q$, which is taken to be fixed at $Q = Q_0^{(g)}$. We truncate this basis to only include the ground and a single excited diabatic state denoted as $|g\rangle$ and $|e\rangle$, respectively. In this basis, $\hat{H}_{mol}$, including a single harmonic vibrational mode, can be written as

$$\hat{H}_{mol} = \frac{\hat{P}^2}{2} + \frac{1}{2}\omega_v^2 \hat{Q}^2 + \left(E_{vert} - \omega_v^2 Q_0^{(e)}\hat{Q}\right)|e\rangle\langle e|, \tag{3}$$

where $\hat{P}$ and $\hat{Q}$ are the mass-weighted momentum and position operators of the vibrational mode, respectively, $E_{vert} = \hbar\Omega_{e,g} + S\hbar\omega_v$ is the energy of the vertical transition from the ground to excited electronic PES at the point of the ground-state nuclear equilibrium configuration (the FC point) with $\Omega_{e,g}$ being the transition frequency between the ground and excited electronic states at their respective nuclear equilibrium position. Furthermore, in Eq. (3) we have taken the ground-state nuclear equilibrium configuration to be at the origin so that $Q_0^{(g)} = 0$, and it is assumed that there is no coupling between diabatic states. We have also assumed that the vibrational frequency is independent of the electronic state.

Equations (1) and (3) effectively describe a system of coupled harmonic oscillators: the vibrational mode and the cavity, which are coupled through an interaction of the cavity field with the molecular dipole. Expanding the quadratic term in Eq. (1) gives

$$\hat{H}_{system} = \hat{H}_{mol} + \hat{H}_{cav} + \hat{H}_{int}, \tag{4}$$

where

$$\hat{H}_{cav} = \frac{\hat{p}_c^2}{2} + \frac{\omega_c^2}{2}\hat{q}_c^2 \tag{5}$$

and

$$\hat{H}_{int} = \omega_c^2 \mathbf{A} \cdot \hat{\boldsymbol{\mu}}\,\hat{q}_c + \frac{\omega_c^2}{2}(\mathbf{A} \cdot \hat{\boldsymbol{\mu}})^2. \tag{6}$$

As is commonly done with Jaynes–Cummings-like models[13,21,52], we ignore the 2nd term of $\hat{H}_{int}$, which is the dipole self energy, reserving a survey of its effects to a follow up study (as it may become significant in ultrastrong coupling[51]). A comparison of the results with and without the dipole self energy are made in Section 4 of the SI.

Once $\hat{H}_{system}$ is expressed in the diabatic basis, each element of the dipole operator in this basis is expanded to the first order with respect to $\hat{Q}$ about the ground-state nuclear equilibrium position ($Q_0^{(g)} = 0$) in order to account for linear coupling between the cavity and the vibrational mode. This gives rise to two cavity-vibration interaction terms,

$$\hat{H}_{int} = \hat{H}_{PFS} + \hat{H}_{cav-vib}, \tag{7}$$

where

$$\hat{H}_{PFS} = \omega_c^2 \mathbf{A} \cdot \boldsymbol{\mu}_e^0\,\hat{q}_c, \tag{8}$$

and where

$$\hat{H}_{cav-vib} = \omega_c^2 \mathbf{A} \cdot \boldsymbol{\mu}_e'\hat{Q}\hat{q}_c. \tag{9}$$

Here, $\boldsymbol{\mu}_e^0$ is the permanent dipole moment of the excited state at $Q_0^{(g)}$ and $\boldsymbol{\mu}_e'$ is the derivative of the nuclear dipole with respect to $Q$ (see "Methods"). The term $\hat{H}_{PFS}$ polarizes the bare cavity Fock states (hence, denoted PFS for polarizes Fock states)[53], and its effect on the ESVP-mediated downconversion process is discussed in Section 3 of the SI. The term $\hat{H}_{cav-vib}$ is the dominant interaction term and mixes cavity

and vibrational DOF. Therefore, we only consider $\hat{H}_{system} = \hat{H}_{mol} + \hat{H}_{cav} + \hat{H}_{cav-vib}$.

For the application of the TWA, we keep $\hat{H}_{system}$ in terms of phase space operators; nonetheless, we believe it valuable to connect our model with the coupling constant ($g$) used in Jaynes–Cummings like models[13,21,52]. This is done by substituting $\hat{Q} = \sqrt{\frac{\hbar}{2\omega_v}}(\hat{b} + \hat{b}^\dagger)$ and $\hat{q}_c = \sqrt{\frac{\hbar}{2\omega_c}}(\hat{a} + \hat{a}^\dagger)$ into Eq. (9), as well as taking the rotating wave approximation (RWA), giving rise to the expression

$$g = \omega_c A_0 \boldsymbol{\epsilon}_c \cdot \boldsymbol{\mu}_{IR}^{(e)}, \tag{10}$$

where

$$\boldsymbol{\mu}_{IR}^{(e)} = \boldsymbol{\mu}_e'\sqrt{\frac{\hbar}{2\omega_v}} \tag{11}$$

is the vibrational transition dipole moment, which can be found through electronic structure calculations or from experimental IR spectra[54,55]. Equation (9) can be re-written in terms of $g$ as

$$\hat{H}_{cav-vib} = \frac{2g}{\hbar}\sqrt{\omega_c\omega_v}\,\hat{Q}\,\hat{q}_c. \tag{12}$$

## Relevant molecular parameters

Using the model Hamiltonian described in the "Theoretical model", we determine the relevant molecular properties that are required for the photonic downconversion to occur through ESVP. This is achieved by monitoring the cavity photon occupancy ($\hat{N}_{cav}$), which is representative of the amount of photons that can be harnessed as IR emission if the cavity is leaky (finite $\mathcal{Q}$). Rather than representing the Hilbert space by a truncated basis of cavity photon and vibrational modes, we apply the TWA to arrive at an exact analytical expression for $\langle \hat{N}_{cav}\rangle$ (see SI Section 2 for more details). Note that the RWA is not invoked here. To apply the TWA, we rotate $\hat{H}_{system}$ into a basis of two uncoupled oscillators called the polariton normal mode basis (see Fig. 2 and "Methods"). Due to the displacement of the excited-state PES, upon a vertical excitation of the molecule both the upper and lower polariton harmonic oscillators are displaced along their respective coordinates, as depicted in Fig. 2. This displacement of excited-state polariton oscillators creates a nonequilibrium initial condition that drives the dynamics. At finite temperature, the resulting photon occupancy, following the UV/vis excitation, is given by

$$\langle \hat{N}_{cav}(t)\rangle = \frac{S\omega_v^3}{2(C^2+1)}\left[\frac{1}{2\omega_c}\left(\frac{\sin(\Omega_- t)}{\Omega_-} - \frac{\sin(\Omega_+ t)}{\Omega_+}\right)^2 \right.$$
$$\left. + 2\omega_c\left(\frac{\sin^2(\Omega_- t/2)}{\Omega_-^2} - \frac{\sin^2(\Omega_+ t/2)}{\Omega_+^2}\right)^2\right] + N_\beta, \tag{13}$$

where $C = \frac{\omega_v^2 - \omega_c^2}{4\frac{g}{\hbar}\sqrt{\omega_v\omega_c}}$, and where the polariton frequencies are given by $\Omega_\pm = \sqrt{A \pm B}$ with $A = \frac{1}{2}(\omega_v^2 + \omega_c^2)$ and $B = \frac{1}{2}\sqrt{(\omega_v^2 - \omega_c^2)^2 + 16\frac{g^2}{\hbar^2}\omega_v\omega_c}$. The effect of finite temperature is captured by $N_\beta = \frac{1}{4}\sum_\gamma (c_c^{(\gamma)2}\left(\frac{\Omega_\gamma}{\omega_c} + \frac{\omega_c}{\Omega_\gamma}\right)\coth(\beta\hbar\Omega_\gamma/2)) - \frac{1}{2}$, which describes the thermal distribution of vibrational polaritons on the electronic ground state, before electronic excitation of the molecule, where $\gamma$ denotes the upper ($\gamma = +$) or lower ($\gamma = -$) polariton mode, $\beta$ is the inverse temperature, and $c_c^{(\gamma)}$ is the projection of the cavity mode onto the $\gamma$ polariton mode. In the following analysis, we take the zero temperature limit ($\beta \to \infty$) and assume the cavity to be in perfect resonance with the vibration so that $\omega_c = \omega_v$, yielding $c_c^{(\gamma)} = 1/\sqrt{2}$ for $\gamma = +$ and $\gamma = -$. This results in $N_\beta = \frac{1}{8}\sum_\gamma (\frac{\Omega_\gamma}{\omega_c} + \frac{\omega_c}{\Omega_\gamma}) - \frac{1}{2}$. For weak coupling, where $\Omega_\gamma \approx \omega_c = \omega_v$,

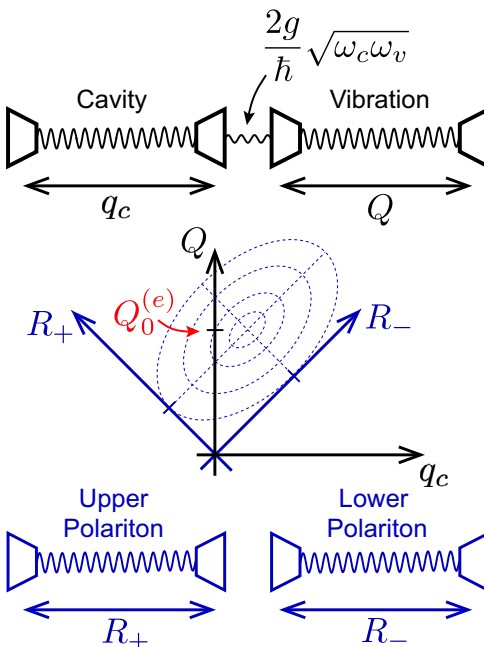

**Fig. 2 | Pictorial representation of the normal mode decomposition for an optical cavity with frequency $\omega_c$ coupled to a molecular vibration with frequency $\omega_v$.** The cavity and vibration harmonic oscillators are depicted as springs. The cavity, with position coordinate $q_c$, and vibration, with position coordinate $Q$, are coupled by $2g\sqrt{\omega_c\omega_v}/\hbar$ where $g$ is the Jaynes–Cummings coupling constant and $\hbar$ is Planck's reduced constant. The cavity and vibrational coordinates are then rotated into a coordinate system of two uncoupled oscillators called the polariton basis, with coordinates $R_-$ and $R_+$, and with displaced minima proportional to the excited-state nuclear equilibrium configuration $Q_0^{(e)}$.

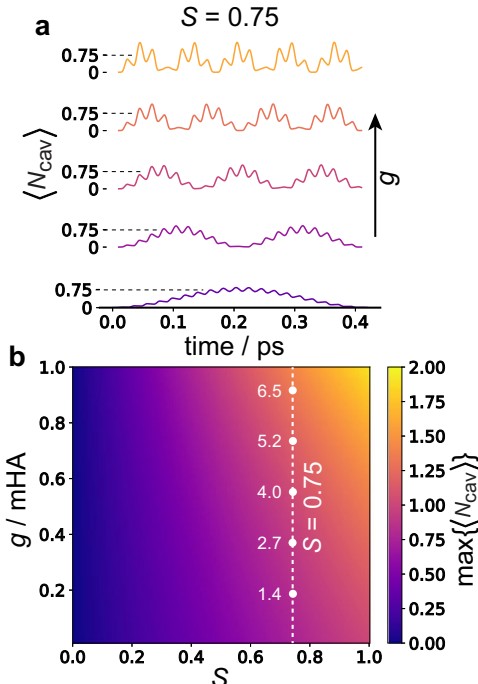

**Fig. 3 | Dependence of the cavity photon number on the Huang–Rhys factor ($S$) and cavity-vibration coupling constant ($g$). a** Time evolution of the cavity photon occupancy ($\langle \hat{N}_{\text{cav}} \rangle$) for a single intramolecular vibration coupled to a cavity following an electronic excitation of the molecule, for different values of $g$ and for a fixed $S$ equal to 0.75. **b** The maximum value of the $\langle \hat{N}_{\text{cav}} \rangle$ time trace (Max$\{\langle \hat{N}_{\text{cav}} \rangle\}$) as a function of both $g$ and $S$. The white dashed line represents $S = 0.75$, with each point on the line corresponding to the $S$ and $g$ values used for the traces in (**a**). The cavity frequency ($\omega_c$) was set in resonance with the vibration frequency ($\omega_v$): $\omega_c = \omega_v = 1600 \text{ cm}^{-1}$.

it is apparent that $N_\beta \approx 0$. But, larger coupling strengths result in $N_\beta > 0$, which is concurrent with the breakdown of the RWA. In that case, the polaritonic ground state within the RWA, which has no photons or vibrational quanta, mixes with the state containing a single cavity photon and single vibration, and therefore, even at zero temperature there will be a nonzero number of cavity photons if the coupling is large.

A key result from Eq. (13) is that $\langle \hat{N}_{\text{cav}}(t) \rangle$ is proportional to $S$. This can be understood by considering that $S$ is proportional to the squared displacement of the excited-state PES. If the displacement is zero, all FC factors will be zero indicating that no vibrational transitions will be induced upon the UV/vis excitation. Increasing $S$ increases the displacement, and consequentially, the number of vibrational quanta created upon electronic excitation. Given a nonzero $g$ value, these vibrational quanta will be converted into cavity photons.

Figure 3a shows the time evolution of $\langle \hat{N}_{\text{cav}} \rangle$ given by Eq. (13) for a cavity mode in perfect resonance with the vibrational mode, i.e., $\omega_c = \omega_v = 7.3$ mHa or 1600 cm$^{-1}$. Results are shown for five evenly spaced $g$ values between 0 and 0.9 mHa. At time zero, $\langle \hat{N}_{\text{cav}} \rangle \approx 0$ for all values of $g$ shown since the UV/vis excitation acts exclusively on the electronic DOF, inducing vibrations on the excited state. Note that there is a small but nonzero number of cavity photons at time zero due to the effect of counter-rotating terms captured in $N_\beta$. This vibrational wavepacket has zero projection onto the cavity mode at time zero and is not an eigenstate of $\hat{H}_{\text{system}}$. As time evolves, Rabi oscillations occur as the wavepacket gains a nonzero projection onto the cavity mode giving rise to the $\langle \hat{N}_{\text{cav}} \rangle$ traces observed. The multi-oscillatory behavior observed in the $\langle \hat{N}_{\text{cav}} \rangle$ traces is due to interference between the contributing polariton frequencies. It can be seen that as $g$ increases so does the maximum value $\langle \hat{N}_{\text{cav}} \rangle$ can have (Max$\{\langle \hat{N}_{\text{cav}} \rangle\}$). For small $g$ values, Max$\{\langle \hat{N}_{\text{cav}} \rangle\}$ is approximately equal to the expected number of vibrational quanta produced upon electronic excitation, $S$. However,

the energy stored in the cavity-vibration coupling will also contribute to $\langle \hat{N}_{\text{cav}} \rangle$, and as $g$ becomes less negligible in comparison to the cavity and vibration energies, $\langle \hat{N}_{\text{cav}} \rangle$ will become noticeably greater than $S$, as observed in Fig. 3a.

Figure 3b shows a heatmap of Max$\{\langle \hat{N}_{\text{cav}} \rangle\}$ as a function of $g$ and $S$. From this, it is clear that maximizing both $g$ and $S$ will lead to the largest Max$\{\langle \hat{N}_{\text{cav}} \rangle\}$ value. Accordingly, a molecule that has a vibrational mode with a significant $\mu_{\text{IR}}^{(e)}$ value, and hence a large $g$ value (see Eq. (10)), along with having significant vibronic coupling, i.e., large $S$, will make a good candidate for use in ESVP-mediated photonic downconversion.

### Radiative and nonradiative dissipation of ESVP

So far our analysis has assumed the molecule and cavity to be fully isolated, apart from their mutual coupling, without including emission from the cavity, which is an integral step of the ESVP-mediated photonic downconversion process. Moreover, practical implementations of this process will suffer from nonradiative relaxation of the intramolecular vibration. It is therefore necessary to consider photonic emission and nonradiative dissipation from the cavity-vibration system. To this end, we are interested in how the photonic emission can be maximized throughout the parameter space of the system. It is evident that a large $S$ value will increase the probability of photonic downconversion as this will produce more excited-state vibrational quanta. Other parameters, however, such as the nonradiative relaxation rate, the cavity-vibration coupling, and the cavity emission rate (related to $\mathcal{Q}$) have a nontrivial relationship to the percentage of incoming UV/vis photons converted to emitted IR photons.

To study the influence of these parameters on the photonic downconversion process, we add two harmonic baths to our model: (1) a bath representing an external electromagnetic field that accounts for

cavity emission, and (2) a bath of (solvent) phonon modes that account for nonradiative vibrational relaxation. Accordingly, the total system-bath Hamiltonian is given by

$$\hat{H}_{\text{total}} = \hat{H}_{\text{system}} + \sum_{i=1}^{E+B}\left(\frac{\hat{p}_i^2}{2} + \omega_i^2\frac{\hat{r}_i^2}{2}\right) + \sum_{i=1}^{E}\kappa_i\,\hat{r}_i\hat{q}_c + \sum_{i=E+1}^{E+B}\chi_i\,\hat{r}_i\hat{Q}, \quad (14)$$

where $\hat{r}_i$ and $\hat{p}_i$ are the position and momentum operators, respectively, for the $1 \le i \le E$ external field modes and $E < i \le E + B$ nonradiative bath modes, $\kappa_i$ is the coupling constant between the $i$th external field mode and the cavity, and $\chi_i$ is the coupling constant between the $i$th nonradiative bath mode and the vibration. It should be noted that, from Eq. (27) and Eq. (32) in "Methods",

$$\kappa_i = \left(\frac{2}{\hbar}\sqrt{\omega_c\omega_i}\right)\sqrt{\frac{\omega_i}{2\pi\mathcal{Q}D_E}}, \quad (15)$$

where $D_E$ is the density of states of the external electromagnetic field, chosen to be constant as a function of $\omega_i$[56]. Hence, the cavity-external field coupling, $\kappa_i$, and $\mathcal{Q}$ are inversely related. This is an intuitive result because as $\mathcal{Q}$ decreases, cavity photons are expected to dissipate more efficiently due to a stronger coupling of the cavity to the external field.

The schematic in Fig. 4a represents the total Hamiltonian given in Eq. (14). All components of the ensemble are comprised of harmonic oscillators, and the red arrows show the coupling between components. Following a similar procedure as was done for the calculation of $\langle\hat{N}_{\text{cav}}\rangle$ in the isolated cavity-vibration case, we used the TWA to explicitly describe the time evolution of the system, nonradiative bath, and external field. This time, normal modes of the $\hat{H}_{\text{total}}$ Hessian matrix were determined. The time evolution was monitored by means of the occupation numbers of the different components.

Figure 4b–d shows the resulting occupation numbers following a UV/vis excitation of the molecule for different $\mathcal{Q}$ values and for a fixed $g$ value of 0.04 mHa. These data show that by changing $\mathcal{Q}$, regimes with quantitatively different behaviors are obtained. For the relatively small value $\mathcal{Q} = 9$ (Fig. 4b), the cavity occupancy is significantly dampened due the strong coupling to the external field. Therefore, the cavity dissipation lifetime is shorter than the Rabi oscillation period. It can also be seen that the nonradiative relaxation (green) outcompetes photonic emission (orange) for this case. For the value $\mathcal{Q} = 100$

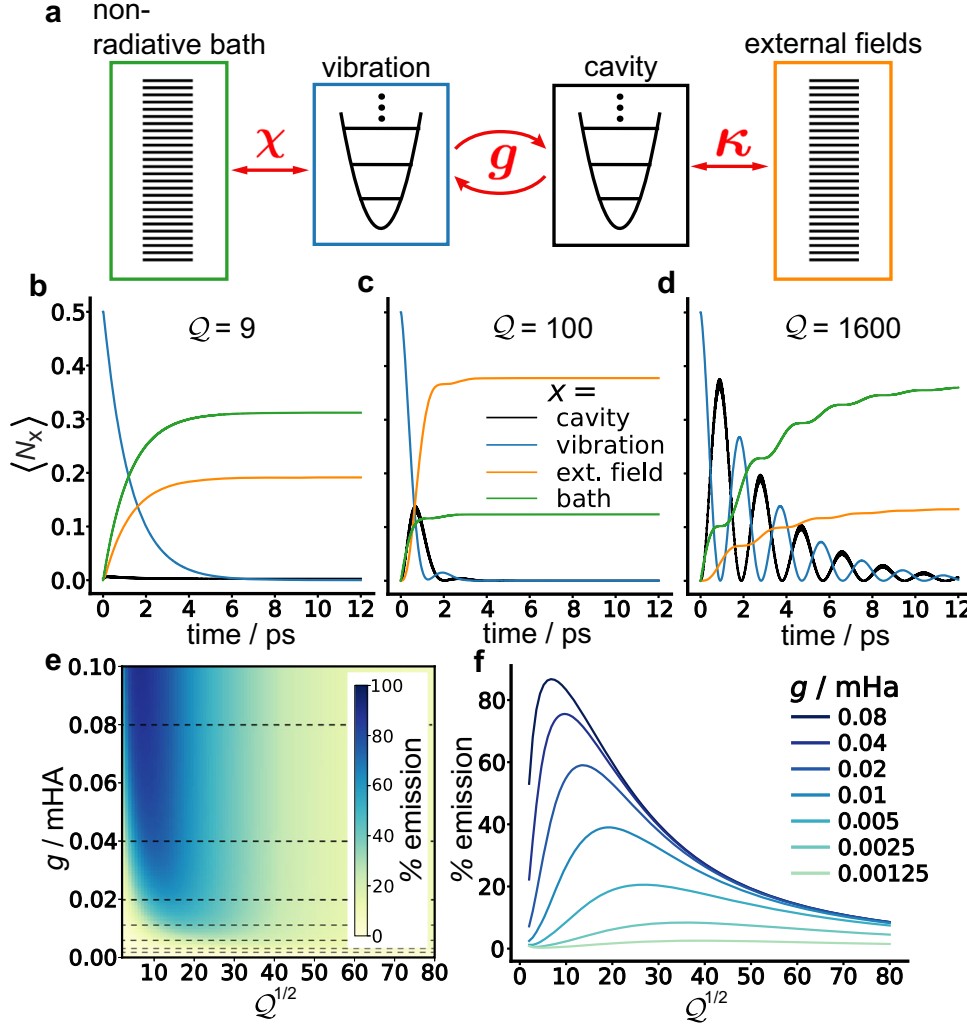

**Fig. 4 | Calculated dynamics, including radiative and nonradiative baths.**
**a** Schematic showing the total Hamiltonian ($\hat{H}_{\text{total}}$) including the cavity–molecule system, the nonradiative bath modes, and the external field modes that constitute emission. **b**–**d** Dynamics of different occupation numbers ($\langle\hat{N}_x\rangle$, where x represents the different subsystems) for different quality factor values ($\mathcal{Q}$) and a coupling constant ($g$) of 0.04 mHa: Black = cavity photons; blue = intramolecular vibrations; orange = external field photons; green = nonradiative bath. **e** A heatmap of the % emission of IR photons as a function of $g$ and $\mathcal{Q}^{1/2}$, where % emission is evaluated at 18 ps. **f** Percent emission as a function of $\mathcal{Q}^{1/2}$ evaluated at different $g$ values. These traces correspond to % emission along the dashed lines in (**e**). The labeled points along the $g = 0.04$ mHa trace correspond to the $\mathcal{Q}$ and $g$ values of the dynamics shown in (**b**–**d**). The cavity frequency ($\omega_c$) was set in resonance with the vibration frequency ($\omega_v$): $\omega_c = \omega_v = 1600$ cm$^{-1}$.

(Fig. 4c), short lived Rabi oscillations are observed. The presence of Rabi oscillations are indicative of strong coupling between the cavity and the molecular vibrational mode. In this case, the photonic emission outcompetes the nonradiative relaxation. For the value $\mathcal{Q} = 1600$ (Fig. 4d), longer-lived Rabi oscillations are observed between the cavity and the molecular vibration mode. In this case, similar to the $\mathcal{Q} = 9$ case, the nonradiative relaxation outcompetes the photonic emission. The change in dynamic behavior with $\mathcal{Q}$ observed in Fig. 4b–d are the result of $\mathcal{Q}$ affecting the cavity decay lifetime (contingent on both the radiative and nonradiative dissipation rates) relative to $g$. The strong coupling regime, indicated by the presence of Rabi-oscillations between the cavity and vibration as well as peak splitting in the emission spectrum, occurs when the cavity decay lifetime is long compared to the cavity-vibration Rabi oscillation period[57].

Figure 4e shows a heatmap of the percentage of vibrations induced by the UV/vis excitation converted to photonic emission (% emission) as a function of $g$ and $\mathcal{Q}$. The % emission is calculated as the ratio of external field photons at equilibrium normalized to the initial number of vibrations created by the UV/vis excitation, given by $S$, that is,

$$\% \, \text{emission} = \frac{1}{S} \sum_i^E \langle \hat{N}_{\text{ext}}^i (t_{\text{eq}}) \rangle. \tag{16}$$

Here, $\hat{N}_{\text{ext}}^i$ is the $i$th external field occupation number operator and $t_{\text{eq}}$ is the time at which equilibrium is reached, here taken to be $t_{\text{eq}} = 18$ ps, which suffices for all values of $g$ and $\mathcal{Q}$ used. By using this definition the % emission is independent of $S$. For example, the results in Fig. 4e were computed with $S = 0.5$ but would be identical for any $S$ value used. Determining the net amount of photons produced in the photonic downconversion process simply requires multiplying the % emission by $S$.

Figure 4f shows slices of the % emission heatmap along $\mathcal{Q}$ for different $g$ values given by the dashed lines in Fig. 4e. The labeled points in 4f indicate the related dynamics in Fig. 4b–d that give rise to those % emission values. It can be seen from Fig. 4f that the % emission increases with $g$ for all values of $\mathcal{Q}$, which indicates that increasing $g$ will always increase the photonic downconversion yield. Furthermore, all traces in Fig. 4 have a maximum along $\mathcal{Q}$, which we denote as $\mathcal{Q}^{\text{max}}$. This is due to the competition between radiative and nonradiative decay. For $\mathcal{Q}$ values larger than $\mathcal{Q}^{\text{max}}$, the probability of cavity photons leaking from the cavity into the external field becomes small enough that nonradiative dissipation outcompetes emission. On the other hand, for $\mathcal{Q}$ values smaller than $\mathcal{Q}^{\text{max}}$, the cavity mode is coupled strong enough to the external field modes that it begins to decouple to the vibration, and again, nonradiative dissipation outcompetes emission. In fact, in the limit where $\mathcal{Q}$ becomes very small, the cavity mode mixes into the external field to the extent that the system effectively becomes a vibration interacting with a free-space electromagnetic field, and radiative emission from the vibration is not expected due to the fast nonradiative relaxation.

$\mathcal{Q}^{\text{max}}$ should be applicable for any polariton system where the matter component has a nonradiative relaxation pathway, implying that a finite cavity $\mathcal{Q}$ value is needed to maximize the photonic emission from such systems. This motivates cavity design principles for measuring and utilizing ESVP-mediated photonic downconversion, as well as for any system that uses an optical resonator to enhance emission.

## Calculation of relevant parameters for pyrene and PAA

The identification of molecules that allow for the formation of ESVP through a UV/vis excitation is critical for all potential applications of the ESVP-mediated photonic downconversion process. Following the results of Figs. 3 and 4, molecules must contain vibrational modes that are both FC active (i.e., having nonzero HR factor) and IR active within

the excited state for the ESVP photonic downconversion to occur. According to the rule of mutual exclusion, such modes exist only in molecules lacking inversion symmetry (vide infra). Furthermore, the extent to which the photonic downconversion process can occur in non-centrosymmetric molecules is dependent on the strength of their vibronic couplings and excited-state IR activity. This section focuses on extracting these molecular properties in specific molecules while making a connection to our theoretical model.

We have performed DFT and time-dependent-DFT (TD-DFT) calculations for two molecules: one with and one without inversion symmetry. Figure 5 shows results for both pyrene and 1-pyrene acetic acid (PAA), which are rigid chromophores with clear vibronic progressions due to vibronic coupling. Pyrene is centrosymmetric and should not have vibrational modes that are both IR and Raman active in accordance with the rule of mutual exclusion[58]. This is relevant as Raman activity implies FC activity. In centrosymmetric molecules, all electronic states contain an inversion symmetry such that electronically exciting them will only induce symmetric vibrations, meaning the excited-state PES is displaced only along these symmetric modes. This displacement allows ground-state vibrational overtones to be reached in the second order Raman process making them Raman active[59]. Importantly, since only symmetric modes are induced upon electronic excitation, which are IR inactive, no ESVP should form in centrosymmetric molecules through UV/vis excitation. PAA, on the other hand, has a carboxylic acid functional group that breaks the inversion symmetry of the pyrene moiety and should contain IR and FC active vibrational modes. By comparing these two molecules we can test whether simple violations of the rule of mutual exclusion can serve to design molecules for ESVP-mediated photonic downconversion.

In order to determine the accuracy of the calculations, absorption spectra were calculated and compared to experimental spectra (Fig. 5a, b). These spectra were calculated using the TWA as detailed in other work[60], and are exact within the FC and harmonic approximation. Results were calculated at finite temperature by taking a Boltzmann population on the ground-state vibrations and also include inhomogeneous broadening due to a Gaussian distribution ($\sigma$ denoting the standard deviation) of the electronic transition energy. At room temperature, calculated spectra (red) with $\sigma = 200$ cm$^{-1}$ show excellent qualitative agreement with the experimental spectra (black) indicating that the calculations capture the correct vibronic couplings and normal modes within the molecules. The calculated spectra at 0.1 K and $\sigma = 10$ cm$^{-1}$ remove the majority of the inhomogenous broadening as well as broadening due to transitions from hot ground-state vibrations that occur at room temperature. This gives "the skeleton" of the vibronic progression with peaks corresponding only to transitions from ground vibrational states on the electronic ground state. A renormalization of the DFT-calculated electronic transition energy was applied for our calculations to match the experimental data.

Figure 5c, d shows $S$ (blue) and $\boldsymbol{\mu}_{\text{IR}}^{(e)}$ (green) for the different vibrational modes as a function of each mode's wavenumber for both pyrene and PAA. Although both molecules contain many modes with nonzero $\boldsymbol{\mu}_{\text{IR}}^{(e)}$ values, we only show $\boldsymbol{\mu}_{\text{IR}}^{(e)}$ for modes with nonzero $S$ values as these are the only vibrational modes that are excited by electronic transitions. $S$ values are calculated for ground-state vibrations and $\boldsymbol{\mu}_{\text{IR}}^{(e)}$ values are calculated for excited-state normal modes. Because Duschinsky rotations can occur[61], the overlap squared of each ground-state mode with each excited-state mode was taken in order to match them. In both molecules, there was generally minimal mixing of ground-state normal modes in the excited state, validating our application of the FC approximation. Figure 5c shows that each FC active vibrational mode has a zero $\boldsymbol{\mu}_{\text{IR}}^{(e)}$ value for pyrene, in accordance with

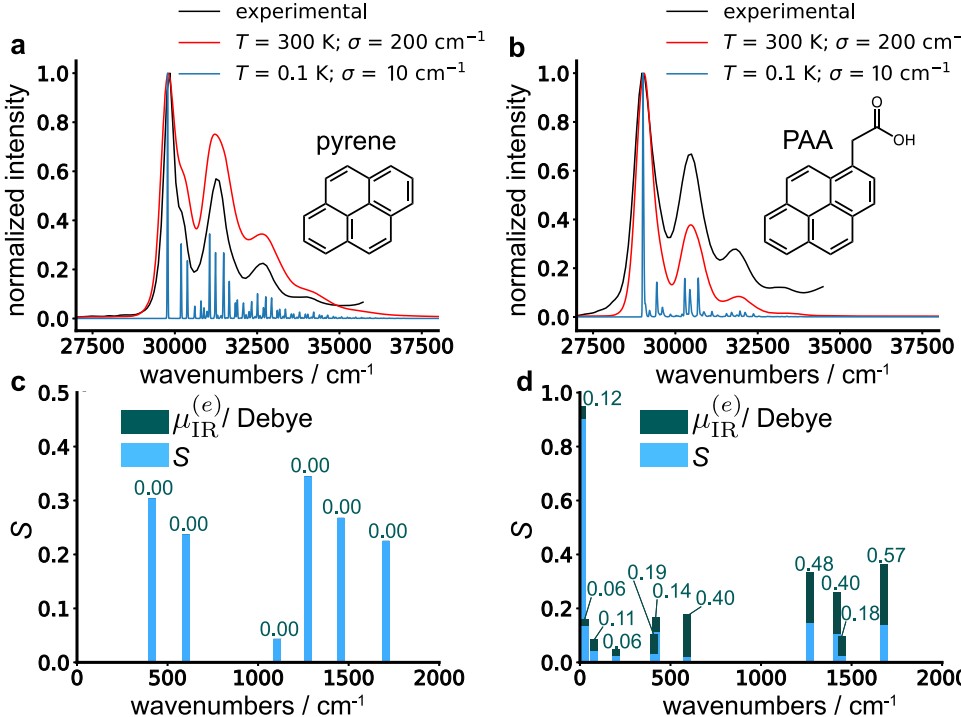

**Fig. 5 | Density functional theory (DFT) calculations of the vibronic couplings and excited-state infrared (IR) dipoles for pyrene and 1-pyrene acetic acid (PAA).** Calculated absorption spectra for (**a**) pyrene and (**b**) PAA were performed at different temperatures (red = 300 K; blue = 0.1 K) and with different standard deviations ($\sigma$) for the Gaussian distribution of electronic energies taken to represent inhomogeneous broadening (red = 200 cm$^{-1}$; blue = 10 cm$^{-1}$). Experimental spectra (black) were taken in cyclohexane for pyrene and toluene for PAA at room temperature (~293 K). Calculated Huang–Rhys factors ($S$) and excited-state IR dipole moments ($\mu_{IR}^{(e)}$) for vibrational modes of (**c**) pyrene and (**d**) PAA are plotted as a function of the wavenumber of each mode. $\mu_{IR}^{(e)}$ is plotted "on top of" the $S$ values, and are only shown for modes with $S > 0$, while its value is indicated above each bar.

the rule of mutual exclusion. On the other hand, Fig. 5d shows that each FC active vibrational mode in PAA has a nonzero $\mu_{IR}^{(e)}$ value due to the broken inversion symmetry of the pyrene moiety. Therefore, an optical cavity may be tuned to one of the FC active vibrational modes of PAA shown in Fig. 5d—a good choice being the highest frequency mode at 1680 cm$^{-1}$—and upon UV/vis excitation of the PAA molecule, ESVP would form.

The results from the DFT calculations presented here can be connected to the theoretical model presented in "Relevant molecular parameters" and "Radiative and nonradiative dissipation of ESVP" through the calculated $S$ and $g$ values. However, for a single molecule coupled to a cavity mode, $g$ is typically very small when compared to values required for the UV/vis-to-IR photonic downconversion process. For example, for the mode at 1680 cm$^{-1}$, $\mu_{IR}^{(e)} = 0.57$ Debye, which corresponds to a $g$ value of $1.8 \times 10^{-6}$ mHa. This is assuming a cavity volume of $V = 1.06 \times 10^{-16}$ m$^3$ using $V = \lambda^3/2$, where $\lambda$ is the wavelength of the cavity mode. Comparing this $g$ value with Fig. 4e, there would be negligible photonic downconversion for all $\mathcal{Q}$ values. The small $g$ value and photonic downconversion yield are due to having only a single molecule coupled to the cavity. Experimental vibrational-polariton measurements rely on an ensemble of molecules to be coupled to a cavity mode, and the Rabi splittings observed correspond to larger coupling strengths than would be expected from each individual molecule[62]. This is refered to as collective strong coupling for which it is known that the Rabi-splitting increases with the square root of the number of cavity-coupled molecules ($N$)[13,63]. As a rough estimate, an effective $g$ value that scales with $\sqrt{N}$ can, therefore, be considered. Using the concentration of molecules present in the optical cavity and knowing the molecule's absorption cross section, one could estimate the number of excited molecules within the cavity and calculate an effective coupling strength from the DFT results. This would allow the % emission from photonic downconversion of the molecule to be

determined as a function of $\mathcal{Q}$. In order to reach a collective coupling strength of 0.04 mHa for the 1680 cm$^{-1}$ mode of PAA within a Fabry–Pérot cavity (which is sufficient for realizing ESVP-mediated photonic downconversion, according to Fig. 4), a concentration of ~8 mM would be required. This was calculated assuming the sample was excited by a 100 fs pulsed laser with a peak power of 500 kW and the number of molecules excited was estimated by the Beer–Lambert law with an approximate extinction coefficient for PAA taken to be $5 \times 10^4$ cm$^{-1}$ M$^{-1}$ and a path length taken as the spacing of the cavity.

Previous works have indicated that single-particle polaritonic Hamiltonians involving large numbers of molecules (referred to as collective strong coupling) give rise to the formation of dark states which may inhibit polaritonic phenomena[13,16], posing an apparent conundrum in view of experimental observations of such phenomena for thermodynamic samples. It should be noted that little is known about the situation for multi-particle polaritonic states, and it is conceivable that increasing the number of particles offsets the detrimental effects due to dark states. The downconversion process proposed here can manifest as a multi-particle phenomenon even for a single molecular excitation, provided that the Huang–Rhys factor is sufficiently large. It is furthermore possible to increase the particles-to-molecules ratio by the decoupling of ground-state vibrations (through state-selective IR dipoles, or by detuning ground-state modes from the cavity through state-selective vibrational frequencies), or by sheer depletion of ground-state molecules by using strongly absorbing molecules and high laser fluences. We reserve an in-depth survey of collective strong coupling to future work. It is also noteworthy that collective strong coupling can be omitted by using plasmonic structures with radically reduced mode volumes and therefore increased cavity-vibration coupling strength[64,65]. Importantly, different from many other polaritonic applications, the presence of small $\mathcal{Q}$ values (due to ohmic losses) in such plasmonic implementations may be

beneficial rather than detrimental for enhancing the yield of the ESVP-mediated downconversion process (cf. Fig. 4f). Although there may be practical complications arising when tuning plasmonic structures to the IR spectral region, previous demonstrations of vibrational–strong coupling with IR active plasmonic structures support the feasibility of this strategy[66–68].

## Discussion

In this work, a UV/vis-to-IR photonic downconversion process is shown to be driven by ESVP for a molecule in an optical cavity. Accordingly, a UV/vis excitation of the molecule induces vibrational polaritons on the molecule's electronic excited-state potential. Such ESVP may then leak IR light, resulting in a photonic downconversion that can potentially be exploited for applications in sensing, quantum information, and in the manipulation of photo-induced reactions. The mechanism and relevant molecular parameters that allow for the downconversion process to occur are the HR factor and the excited-state IR dipole, which both need to be nonzero. According to the rule of mutual exclusion, this implies that the photonic downconversion process can only occur within molecules lacking inversion symmetry.

By studying the parameter space of the ESVP-mediated photonic downconversion while including both radiative and nonradiative dissipation, we determined the optimal parameters to maximize the downconversion yield, facilitating experimental and practical implementations. Due to the inclusion of nonradiative dissipation, the photonic downconversion is maximized at a specific cavity quality factor. With excess quality factors, the probability of cavity photon leakage becomes small and nonradiative processes will out-compete emission. However, with lower than optimal quality factors, the cavity mode becomes highly delocalized through mixing with free-space electromagnetic field modes outside of the cavity inhibiting strong coupling with the molecular vibration. These two limiting behaviors give rise to an intermediate quality factor, $\mathcal{Q}^{\mathrm{max}}$, that maximizes the downconversion process. In this work, we have restricted our analysis to the single-molecule limit. Future work will focus on the role of collective coupling, paying special attention to the emergence of dark states and the role of disorder in the dipole orientations, energetic disorder, and electronic coupling between molecules. It should be noted that these factors are of relevance to polaritonic phenomena in general, and experimental confirmations of such phenomena seem to suggest some level of immunity against such factors, which we plan to explore in detail.

In the previous subsection, we demonstrated how to use DFT calculations to predict if ESVP will form on a molecule when it is excited with UV/vis light within a cavity. We believe this to be useful for designing molecules that maximize the ESVP-mediated photonic downconversion, allowing for applications in quantum information and sensing to be realized. Such DFT calculations can also determine if an optical cavity could be used to modulate a molecule's excited-state dynamics following a UV/vis excitation with ramifications for photo-induced reactions. The importance of the rule of mutual exclusion is emphasized by our evaluation of both pyrene, containing and inversion symmetry, and 1-pyrene acetic acid, lacking an inversion symmetry. Our results show that only 1-pyrene acetic acid has both FC active (nonzero HR factor) and excited-state IR active vibrational modes, which permit the formation of ESVP.

We note that a potential exception to the requirement of molecular inversion symmetry for the formation of ESVP induced by a UV/vis excitation would be the occurrence of excited-state symmetry breaking due to solvent interactions[69]. This results in ground-state vibrational modes of different symmetries mixing in the excited state and could allow excited-state IR active vibrations to be produced on ground-state centrosymmetric molecules following a UV/vis excitation of the molecule. These vibrations could, therefore, couple to a cavity, opening further ways to mediate the photonic downconversion.

## Methods

### Diabatic expansion of the dipole operator

The dipole operator can be separated into components that depend on electronic and nuclear DOF as

$$\hat{\boldsymbol{\mu}} = \hat{\boldsymbol{\mu}}_{\mathrm{el}} + \hat{\boldsymbol{\mu}}_{\mathrm{nu}}. \tag{17}$$

When expanding $\hat{H}_{\mathrm{system}}$ in the diabatic basis, each element of the dipole operator in this basis becomes

$$\hat{\boldsymbol{\mu}}_{\alpha\beta}(\hat{Q}) = \langle\alpha|\hat{\boldsymbol{\mu}}_{\mathrm{el}}|\beta\rangle + \hat{\boldsymbol{\mu}}_{\mathrm{nu}}(\hat{Q})\delta_{\alpha\beta}, \tag{18}$$

which act on the nuclear Hilbert space of $\hat{Q}$ and not on the electronic Hilbert space. Note that $\hat{\boldsymbol{\mu}}_{\mathrm{nu}}$ does not act on the diabatic basis states as they do not depend on $Q$ (see Eq. (2)). The linear expansion of each element of the dipole operator is given by

$$\hat{\boldsymbol{\mu}}_{\alpha\beta}(\hat{Q}) = \boldsymbol{\mu}_{\alpha\beta}\left(Q_0^{(g)}\right) + \left.\frac{\partial\boldsymbol{\mu}_{\alpha\beta}(Q)}{\partial Q}\right|_{Q_0^{(g)}}\left(\hat{Q} - Q_0^{(g)}\right), \tag{19}$$

with the first coefficient being

$$\boldsymbol{\mu}_{\alpha\beta}\left(Q_0^{(g)}\right) = \langle\alpha|\hat{\boldsymbol{\mu}}_{\mathrm{el}}|\beta\rangle + \boldsymbol{\mu}_{\mathrm{nu}}\left(Q_0^{(g)}\right)\delta_{\alpha\beta}, \tag{20}$$

where $\langle\alpha|\hat{\boldsymbol{\mu}}_{\mathrm{el}}|\beta\rangle$ is the permanent electic dipole of state $\alpha$ when $\alpha=\beta$ and the transition dipole between the states $\alpha$ and $\beta$ when $\alpha\neq\beta$. Further, $\boldsymbol{\mu}_{\mathrm{nu}}(Q_0^{(g)})$ is the dipole moment of the nuclei at configuration $Q_0^{(g)}$. It follows that the second coefficient in Eq. (19) is given by the derivative of the nuclear dipole moment with respect to $Q$ at configuration $Q_0^{(g)}$,

$$\left.\frac{\partial\boldsymbol{\mu}_{\alpha\beta}(Q)}{\partial Q}\right|_{Q_0^{(g)}} = \left.\frac{\partial\boldsymbol{\mu}_{\mathrm{nu}}(Q)}{\partial Q}\right|_{Q_0^{(g)}}\delta_{\alpha\beta}. \tag{21}$$

We ignored coupling between the cavity and the molecule's electronic DOF as the cavity mode is assumed to be near resonant with the vibrational mode of interest making it significantly off-resonant from the electronic transition. Accordingly, we set elements of the dipole operator where $\alpha\neq\beta$ to zero. Because of this, the electronic ground and excited-state contributions to $\hat{H}_{\mathrm{system}}$ are uncoupled and can be treated separately. For the purpose of this study, we focused on the excited-state contribution where $\alpha=\beta=e$. To simplify notation we defined

$$\boldsymbol{\mu}'_e \equiv \left.\frac{\partial\boldsymbol{\mu}_{ee}(Q)}{\partial Q}\right|_{Q_0^{(g)}}, \tag{22}$$

and

$$\boldsymbol{\mu}_e^0 \equiv \boldsymbol{\mu}_{ee}\left(Q_0^{(g)}\right) - \boldsymbol{\mu}'_e Q_0^{(g)}. \tag{23}$$

Note that $\boldsymbol{\mu}_e^0$ is the permanent dipole moment of the excited state at the ground-state equilibrium configuration, which was taken to be $Q_0^{(g)}=0$.

### Polariton normal mode basis

$\hat{H}_{\mathrm{system}}$ can be rotated into a basis of two uncoupled oscillators called the polariton normal mode basis (see Fig. 2). This basis is found by determining the eigenvectors of the cavity-vibration Hessian matrix,

$$\mathbf{H} = \begin{pmatrix} \omega_c^2 & G \\ G^* & \omega_v^2 \end{pmatrix}. \tag{24}$$

The polariton harmonic oscillators have upper frequency ($\Omega_+$) and lower frequency ($\Omega_-$) given by the square root of the eigenvalues of $\mathbf{H}$.

The off-diagonal element of the Hessian matrix, $G$, is the cavity-vibration bilinear coupling constant given by

$$G = \frac{2g}{\hbar}\sqrt{\omega_c \omega_v}. \tag{25}$$

The analytical expression for $\langle \hat{N}_{cav}(t)\rangle$ was determined by rotating the cavity photon occupancy operator,

$$\hat{N}_{cav} = \frac{1}{2\hbar\omega_c}(\omega_c^2 \hat{q}_c^2 + \hat{p}_c^2) - 1/2, \tag{26}$$

into the polariton basis by expanding $\hat{q}_c$ and $\hat{p}_c$ in terms of polariton phase space operators, $\hat{R}_-$ and $\hat{R}_+$. The Weyl symbol of $\hat{N}_{cav}$, in the polariton basis, was determined by replacing the polariton phase space operators with phase space variables that can be evolved according to the classical equations of motion for two uncoupled harmonic oscillators. From this, we calculate the expectation value of the time-evolved Weyl symbol of $\hat{N}_{cav}$ acting on the time-independent Wigner distribution of the initial position and momentum of the system[43,70].

### System-bath couplings

An ohmic spectral density was taken for both the nonradiative bath and the external field, $J_E(\omega) = \eta_E \omega$ and $J_B(\omega) = \eta_B \omega$, respectively, where $\eta_E$ ($\eta_B$) is the cavity (vibration) damping constant due to coupling to the external field (nonradiative bath). By taking a large but finite number of nonradiative bath and external field modes (we used 500 modes for each bath), $\hat{H}_{total}$ approximates the thermodynamic number of bath modes under experimental conditions[71]. In our modeling, we used a constant density of states ($D_B$ for the nonradiative bath; $D_E$ for the external field) and frequency-dependent coupling constants

$$\kappa_i' = \sqrt{\frac{\eta_E \omega_i}{D_E}}, \tag{27}$$

and

$$\chi_i' = \sqrt{\frac{\eta_B \omega_i}{D_B}}, \tag{28}$$

which are related to the bilinear coupling constants $\kappa_i$ and $\chi_i$ in Eq. (14) by $\kappa_i = 2\kappa_i'\sqrt{\omega_c\omega_i}/\hbar$ and $\chi_i = 2\chi_i'\sqrt{\omega_c\omega_i}/\hbar$. Here the prime denotes the coupling constants, used in Jaynes–Cummings-like models. In all of our calculations, the vibrational damping constant, $\eta_B$, was fixed such that an excited vibration would have a lifetime of 2 ps for $g = 0$, which is typical of condensed-phase molecular vibrations (we study different nonradiative lifetimes in Section 5 of the SI)[26,27]. Furthermore, $\eta_E$ is related to $\mathcal{Q}$, which is defined as the ratio of the cavity center frequency to its emission linewidth, $\Delta\omega_c$[72],

$$\mathcal{Q} = \frac{\omega_c}{\Delta\omega_c}. \tag{29}$$

An expression for the emission profile linewidth of an optical cavity coupled to a bath of external field modes can be extracted from the Lindblad master equation as[45]

$$\Delta\omega_c = 2\pi D_E |\kappa'_{\omega_c}|^2, \tag{30}$$

where $\kappa_{\omega_c}$ is the cavity-external field coupling at the cavity frequency. Substituting Eq. (30) into Eq. (29) gives

$$\mathcal{Q} = \frac{\omega_c}{2\pi D_E |\kappa'_{\omega_c}|^2}. \tag{31}$$

Setting $\kappa_i' = \kappa'_{\omega_c}$ and $\omega_i = \omega_c$ in Eq. (27) and then comparing the result to Eq. (31), it follows that

$$\eta_E = \frac{1}{2\pi\mathcal{Q}}. \tag{32}$$

It should be noted that the Lindblad equation invokes the RWA and therefore will give inaccurate $\mathcal{Q}$ values in the ultrastrong cavity-external field coupling regime.

### DFT calculations of Huang–Rhys factors and excited-state IR dipoles

DFT calculations were performed using the Qchem software (version 5.3). All DFT and TD-DFT calculations utilized the B3LYP functional and 6−31+G* basis set. In order to determine $S$ and $\mu_{IR}^{(e)}$ for pyrene and PAA, first the ground-state equilibrium geometry was calculated for each molecule in its singlet ground state, followed by a normal mode analysis about this ground-state minimum to obtain the intramolecular vibrations, both done via DFT calculations. TD-DFT calculations were then used to determine the forces felt by the nuclei due to the molecule being excited into the first, bright excited singlet state. This force is due to the electron density of the excited state acting on the ground-state nuclear configuration, shifting the equilibrium nuclear configuration. Excited-state force vectors from the TD-DFT calculations were projected onto the ground-state vibrational modes. These forces determine the shift in the excited-state PES and were used to calculate HR factors for each vibrational mode. Finally, we performed TD-DFT excited-state normal mode analyses at the FC point to calculate the excited state intramolecular vibrations and their corresponding $\mu_{IR}^{(e)}$ values. No description of solvent interactions was included in the computed spectra.

### Calculation of Huang–Rhys factors and absorption spectra

Here we provide details on the calculation of the absorption spectra in Fig. 5. Let $\mathbf{r}$ and $\mathbf{p}$ be the nuclear position and momentum vectors in the Cartesian space of a molecule (or in any arbitrary orthonormal basis) with $N$ nuclei. In the diabatic basis, the molecular Hamiltonian becomes

$$\hat{H}(\mathbf{r},\mathbf{p}) = \frac{\mathbf{p}^2}{2}\hat{\mathbb{1}} + \sum_{\alpha\beta} h_{\alpha\beta}(\mathbf{r})|\alpha\rangle\langle\beta|, \tag{33}$$

where the elements $h_{\alpha\beta}$ with $\alpha = \beta$ (diagonal elements) are the diabatic PES and the elements with $\alpha \neq \beta$ (off-diagonal elements) couple these surfaces at position $\mathbf{r}$. Taking the approximation that the off-diagonal elements are zero and assuming the diagonal diabatic PES are harmonic by expanding them to the second order about a reference configuration, $\mathbf{r_0}$, Eq. (33) becomes

$$\hat{H} = \frac{\mathbf{p}^2}{2}\hat{\mathbb{1}}$$
$$+ \sum_{\alpha}\left[h_\alpha(\mathbf{r_0}) + \nabla h_\alpha(\mathbf{r})|_{\mathbf{r_0}}(\mathbf{r} - \mathbf{r_0}) + \frac{1}{2}(\mathbf{r} - \mathbf{r_0})^T \mathbf{H}_{mol}^\alpha(\mathbf{r} - \mathbf{r_0})\right]|\alpha\rangle\langle\alpha|. \tag{34}$$

The Hessian matrix, $\mathbf{H}_{mol}^\alpha$, is given by

$$\mathbf{H}_{mol}^\alpha = \frac{\partial^2 h_\alpha}{\partial r_i \partial r_j}, \tag{35}$$

where $r_i$ and $r_j$ are the $i$th and $j$th element of $\mathbf{r}$, respectively.

For the calculation of absorption spectra, we only consider the ground and a single excited diabatic state, typically the first bright excited singlet state. We make the Franck–Condon approximation by

assuming that the ground and excited-state Hessian matrices are the same, i.e., $\mathbf{H}_{mol}^g = \mathbf{H}_{mol}^e$, and that they are determined by a vibrational analysis on the electronic ground state. That is, the ground-state normal modes are used for the excited-state diabatic PES, which are displaced from the ground-state PES by an amount related to the vibronic coupling. Taking $\mathbf{r_0}$ to be the ground-state equilibrium configuration, and defining our coordinate system such that $\mathbf{r_0} = 0$, Eq. (34) becomes

$$\hat{H} = \frac{\mathbf{p}^2}{2}\hat{\mathbb{1}} + \left(E_g + \frac{1}{2}\mathbf{r}^T\mathbf{H}_{mol}^g\mathbf{r}\right)|g\rangle\langle g| + \left(E_e - \mathbf{f}\cdot\mathbf{r} + \frac{1}{2}\mathbf{r}^T\mathbf{H}_{mol}^g\mathbf{r}\right)|e\rangle\langle e|. \tag{36}$$

Here, $E_g$ and $E_e$ are the energy of the ground and excited-state diabatic PES at the ground-state reference configuration $\mathbf{r_0}$, respectively, and $\mathbf{f}$ is the force vector that the excited-state electronic structure exerts on the nuclei at $\mathbf{r_0}$. Equation (36) is then transformed into the normal mode basis,

$$\hat{H} = \left(\sum_i^{3N}\frac{\hat{p}_i^2}{2}\right)\hat{\mathbb{1}} + \left(E_g + \sum_i^{3N}\frac{\omega_i^2\hat{q}_i^2}{2}\right)|g\rangle\langle g| + \left(E_e + \sum_i^{3N}(-\tilde{f}_i)\hat{q}_i + \frac{\omega_i^2\hat{q}_i^2}{2}\right)|e\rangle\langle e|, \tag{37}$$

where $\hat{q}_i$ and $\hat{p}_i$ are the position and momentum operators, respectively, along the $i$th eigenvector (i.e., normal mode) of $\mathbf{H}_{mol}^g$ with eigenvalue $\omega_i^2$, and $\tilde{f}_i$ is the projection of $\mathbf{f}$ along the $i$th normal mode. The excited-state force, $\tilde{f}_i$, is related to the displacement ($d_i$) between the ground and excited-state PES minimum along the $i$th normal mode by

$$\tilde{f}_i = \omega_i^2 d_i, \tag{38}$$

and to the Huang–Rhys factor of the $i$th normal mode ($S_i$) by

$$S_i = \frac{\tilde{f}_i^2}{2\hbar\omega_i^3}. \tag{39}$$

Using Eqs. (38) and (39), Eq. (37) can be recast as

$$\hat{H}(\mathbf{r},\mathbf{p}) = \sum_i^N \frac{\hat{p}_i^2}{2}\hat{\mathbb{1}} + \frac{\omega_i^2\hat{q}_i^2}{2}\hat{\mathbb{1}} + E_g|g\rangle\langle g| + \left(E_e - \sum_i^N \omega_i^2 d_i\hat{q}_i\right)|e\rangle\langle e|, \tag{40}$$

which is the form of the molecular Hamiltonian in Eq. (3) once the position and momentum variables have been promoted to operators and a single normal mode is selected. Also, $E_g$ can be taken as the reference energy and set to zero.

Based on this diabatic formulation of the molecular Hamiltonian, electronic structure calculations can be used to determine the parameters necessary for calculating the absorption spectra, which are the normal modes and frequencies from $\mathbf{H}_{mol}^g$ as well as their respective Huang–Rhys factors. This is done by: (1) Calculating the optimized ground-state geometry of the molecule, which will be taken as the reference geometry, $\mathbf{r_0}$. (2) Performing a normal mode analysis about $\mathbf{r_0}$ to obtain the mass-weighted Hessian matrix $\mathbf{H}_{mol}^g$, which is used to find the mass-weighted normal modes and their respective eigenvalues ($\omega_i^2$). (3) Calculating the excited-state forces acting on the ground-state equilibrium geometry, $\mathbf{r_0}$, to determine $\mathbf{f}$. (4) Projecting $\mathbf{f}$ onto each normal mode to obtain $\tilde{f}_i$ values, which by Eq. (39) allows for the calculation of the $S_i$ values for each normal mode. Once these parameters are found, the following Eq. gives the response function for the linear absorption process at any finite temperature within the Franck–Condon and harmonic approximations[60]

$$I(t) = e^{-\frac{i}{\hbar}\left(E_0 t - (\sigma_{E_0}t)^2/2\right)}\, e^{-\sum_i 2\frac{\omega_i S_i}{\hbar}\sigma_{\hat{q}_i}^2}\, e^{\sum_i S_i\left(\left(\sigma_{\hat{q}_i}\sigma_{\hat{p}_i} - \frac{1}{2}\right)e^{i\omega_i t} + \left(\sigma_{\hat{q}_i}\sigma_{\hat{p}_i} + \frac{1}{2}\right)e^{-i\omega_i t}\right)}, \tag{41}$$

where

$$\sigma_{\hat{q}_i} = \sqrt{\frac{\hbar}{2\omega_i\tanh\left(\frac{\beta\hbar\omega_i}{2}\right)}} \tag{42}$$

$$\sigma_{\hat{p}_i} = \omega_i\sigma_{\hat{q}_i}, \tag{43}$$

and where $\sigma_{E_0}$ is the inhomogeneous broadening linewidth of the electronic transition.

The absorption power spectrum is calculated by a Fourier transform of the response function

$$A(\omega) = \int_0^\infty I(t)e^{-i2\pi\omega t}dt. \tag{44}$$

The response function of pyrene is given in Section 8 of the SI as an example.

## Reporting summary
Further information on research design is available in the Nature Portfolio Reporting Summary linked to this article.

## Data availability
The data supporting the findings of this study are provided in the source data file and upon request from authors. The density functional theory (DFT) structure, frequency, and force data supporting the plots herein are available on Mendeley data[73]. Source data are provided with this paper.

## Code availability
The code used for calculating time-evolved occupation values, analyzing the DFT outputs, and calculating absorption spectra, along with raw data files for each plot, are available on GitHub[74].

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

## Acknowledgements
The authors thank Jonathan D. Schultz for the helpful discussions regarding this work as well as Rafael Eduardo López-Arteaga for helpful discussions and supplying absorption data of pyrene and 1-pyrene acetic acid. This work was supported as part of the Center for Molecular Quantum Transduction (CMQT), an Energy Frontier Research Center funded by the U.S. Department of Energy, Office of Science, Basic Energy Sciences under Award No. DE-SC0021314. This material is based upon work supported by the National Science Foundation Graduate Research Fellowship under Grant No. DGE-2234667.

## Author contributions
C.K.T.W. performed the calculations, contributed ideas, and wrote the manuscript. J.P. contributed to ideas, derived the analytical expression for the photon occupation number, and edited the manuscript. E.A.W. contributed to ideas and edited the manuscript. R.T. proposed and oversaw the project, contributed ideas, and edited the manuscript.

## Competing interests
The authors declare no competing interests.
