## [Peer Review file · Nature Communications]

REVIEWER COMMENTS

Reviewer #1 (Remarks to the Author):

The paper presents a well-written account of a novel phenomenon in which the vibrational relaxation energy following a vertical FC excitation in an organic chromophore is efficiently converted into IR cavity photons in a tuned infrared cavity. Generally, the suggested scheme can be used to control the fate of excited state processes such as the excimer formation and single exciton fission which depend on vibronic coupling, and should therefore generate significant interest in a broad community of researchers working on exploiting organic materials for technological purposes.

The authors analyze the effect theoretically using a Jaynes-Cummings type Hamiltonian, showing how an optimal choice of the coupling parameter g and the cavity quality factor Q can result in a near complete conversion of the initial vibrational energy following impulsive excitation into cavity photons. The authors also analyze candidate chromophores based on perylene which should produce the effect. The paper is clearly written, logically developed and the results are of sufficient novelty and importance to justify publication.

Perhaps the main criticism, already noted by the authors, is that the analysis applies to a single chromophore, where the associated coupling constant g is entirely too small to observe the effect. Large ensembles of N molecules coupled to the cavity mode are required, as the coupling scales as $N^{1/2}$. This however leads to additional concerns - for example, the impact of orientational disorder with respect to the transition dipole moments, the presence of generalized energetic disorder and the likely electronic coupling between molecules. The authors should discuss such realistic concerns and at least estimate the concentration and volume needed to observe the effect with the suggested perylene chromophores.

One more technical issue concerns the Hamiltonian. The authors should write down explicitly the interaction term in the Hamiltonian which includes g .

Reviewer #2 (Remarks to the Author):

Vibrational strong coupling has attracted much attention as a means to manipulate potential energy surfaces of materials. Understanding the property of hybridized states formed from the interaction has been challenging because of the complexity of the system. C. K. Terry Weatherly et al. used truncated Wigner approximation (TWA) to predict the dynamics of excited-state vibrational polariton that have

remained unexplored. The TWA allowed the inclusion of non-radiative baths and external fields at an affordable cost, and the authors successfully simulated the photonic down conversion yield under strong coupling, which would have a significant contribution to the polariton research field. The calculations showed that polariton formation can enhance the IR emission process which can be maximized at some cavity quality factor. The authors also discuss the possible condition to obtain the enhancement in the case of pyrene derivatives. The manuscript and supporting information are carefully written in a manner that experimentalists can follow straightforwardly. I believe the paper will be well-received by the broad community in physics and chemistry working on polariton. I recommend the publication of this manuscript with minor corrections.

Here are my comments to the authors.

(1) Main text, Page 14 line 26: The authors mentioned the contribution of collective effect to the coupling constant (g). Use of plasmonic structures with small mode volume can also increase the interaction. I suggest authors include descriptions and citations to this point.

(2) It might be kinder to put subscript “c” for p and q used for the cavity (ex. p_c and q_c as seen in the Supporting Information, Page 9 Line 4) since authors use them both for cavity and polariton in the Supporting Information.

(3) In Figure S1b, what is the origin of the region $\langle N_{cav} \rangle > 2.00$ shown in yellow?

Reviewer #3 (Remarks to the Author):

The article by Weatherly et al. proposes a hypothetical photonic down conversion process where UV/vis photons are converted into IR radiation via polariton modes generated from the Franck-Condon electronic excitation process of a molecule in an infrared microcavity in the regime of vibrational strong light-matter coupling.

This a timely topic and the proposed idea is new as far as I am aware. However, there are a few points of the theoretical treatment that are unclear, and almost no attention has been paid to some key features of polaritonic systems that would likely make the feasibility of the process hypothesized by the authors to be essentially negligible. If the authors can address these points satisfactorily, then I would probably recommend publication at this journal. The specific issues that should be addressed are:

1. The authors have emphasized in their discussion that experiments in the vibrational strong coupling regime occur with a large ensemble of molecules, yet their modeling always considers only a single molecule with unrealistically large values of single-molecule coupling strength. There is no argument put by the authors on whether the efficiencies they compute in Figs 3 and especially 4 are expected to persist when an ensemble of molecules interact with the cavity mode and a large density of dark states result from that. In fact, simple arguments in the literature suggest that any time a large density of molecules is considered effects obtained from single-molecule models such as those reported by the authors become irrelevant. If there is a reason to expect that the downconversion suggested by the authors will not have infinitesimal efficiency under ensemble conditions, the authors should clarify that, and, if there is no reason the proposed effect would ever be even observable, then, this publication is probably better suited for a more specialized journal.

To be clear, based on current literature (e.g., Feist, Subotnik, Yuen-Zhou), the impression one has is that within a single-photon mode description of the cavity, a generalization of the current approach to include many molecules and the resulting dark states would lead to an extremely small efficiency of the downconversion process because the electronic excitation from the ground-state would have much larger probability to excite dark states than polaritons and because polariton energy dissipation would also direct significant amounts of more energy to dark states reducing even more the fraction of IR energy coming from the microcavity. If this argument doesn't apply here, then it would be important to get the authors perspective on these points since these are ideas currently permeating the polariton literature.

2. The authors use the Hamiltonian including the counterrotating-wave terms but ignore the A^2 term on the basis that the latter should only be relevant under ultrastrong coupling. However, the same argument used to ignore A^2 terms is often used to ignore the counterrotating as it happens frequently in models of strong light-matter coupling. Thus, it seems inconsistent to ignore one type of term and not the other. The authors should clarify this. What is the effect of the A^2 term? If the computational cost is not larger, then why not do the calculation with the full Hamiltonian? In fact, I think the full Hamiltonian calculations should be given at least in the Supplementary Material, because if they turn out to lead to significantly different results than the computations without the A^2 term, that could point to an issue that would have to be addressed.

3. At OK the authors mention that regardless of the value of the light-matter coupling, the initial photon population in the cavity is zero. Given that counterrotating terms are included in the Hamiltonian, wouldn't the cavity-matter composite ground-state have a non-zero photon occupation number that also increases with the light-matter coupling? Did the authors ignore this non-zero photon occupation number in the ground-state when calculating $\langle N_{\text{cav}}(t) \rangle$? If yes, then needs to be explained, and if no, then the reason there is zero photonic content in the total ground-state should also be discussed since this does not seem like a generic feature of models including counterrotating terms.

Reviewer 1's comments:

The paper presents a well-written account of a novel phenomenon in which the vibrational relaxation energy following a vertical FC excitation in an organic chromophore is efficiently converted into IR cavity photons in a tuned infrared cavity. Generally, the suggested scheme can be used to control the fate of excited state processes such as the excimer formation and single exciton fission which depend on vibronic coupling, and should therefore generate significant interest in a broad community of researchers working on exploiting organic materials for technological purposes.

The authors analyze the effect theoretically using a Jaynes-Cummings type Hamiltonian, showing how an optimal choice of the coupling parameter g and the cavity quality factor Q can result in a near complete conversion of the initial vibrational energy following impulsive excitation into cavity photons. The authors also analyze candidate chromophores based on perylene which should produce the effect. The paper is clearly written, logically developed and the results are of sufficient novelty and importance to justify publication.

Response: We thank the reviewer for the positive evaluation of our work and detailed reading.

Perhaps the main criticism, already noted by the authors, is that the analysis applies to a single chromophore, where the associated coupling constant g is entirely too small to observe the effect. Large ensembles of N molecules coupled to the cavity mode are required, as the coupling scales as $N^{1/2}$. This however leads to additional concerns - for example, the impact of orientational disorder with respect to the transition dipole moments, the presence of generalized energetic disorder and the likely electronic coupling between molecules. The authors should discuss such realistic concerns and at least estimate the concentration and volume needed to observe the effect with the suggested perylene chromophores.

Response: Indeed, as we noted in our manuscript, our analysis applies to a single chromophore which serves as an important step before introducing complexities associated with large ensembles of molecules. We thank the reviewer for bringing up dipole orientation disorder, energetic disorder, and electronic coupling between molecules, all of which we plan to investigate in detail in upcoming work. The reason we believe our findings are to a large extent immune to these factors is owing to the ground state vibrational polariton literature, which at this point features many experimental reports on vibrational polaritons despite being susceptible to the same factors. We have added a brief discussion of this in the revised manuscript on page 16, lines 429-434. We have also added a crude estimate of the concentration of molecules needed to reach the desired coupling strength on page 15, lines 382-388.

Lastly, we wanted to bring up the point also raised by Reviewer 2 regarding the use of plasmonics to realize the proposed effect by dramatically increasing the mode volume to achieve the required coupling strengths in the single molecule limit. We discuss this in the revised manuscript on page 15, lines 402-407.

One more technical issue concerns the Hamiltonian. The authors should write down explicitly the interaction term in the Hamiltonian which includes g .

Response: We agree that this would be beneficial and the reader is now explicitly shown how the cavity-vibration interaction term of the Hamiltonian can be written in terms of g in Eq. 12 on page 6.

Reviewer 2's comments:

Vibrational strong coupling has attracted much attention as a means to manipulate potential energy surfaces of materials. Understanding the property of hybridized states formed from the interaction has been challenging because of the complexity of the system. C. K. Terry Weatherly et al. used truncated Wigner approximation (TWA) to predict the dynamics of excited-state vibrational polariton that have remained unexplored. The TWA allowed the inclusion of non-radiative baths and external fields at an affordable cost, and the authors successfully simulated the photonic down conversion yield under strong coupling, which would have a significant contribution to the polariton research field. The calculations showed that polari-

ton formation can enhance the IR emission process which can be maximized at some cavity quality factor. The authors also discuss the possible condition to obtain the enhancement in the case of pyrene derivatives. The manuscript and supporting information are carefully written in a manner that experimentalists can follow straightforwardly. I believe the paper will be well-received by the broad community in physics and chemistry working on polariton. I recommend the publication of this manuscript with minor corrections.

Response: We appreciate the detailed review of our work and positive feedback.

(1) Main text, Page 14 line 26: The authors mentioned the contribution of collective effect to the coupling constant (g). Use of plasmonic structures with small mode volume can also increase the interaction. I suggest authors include descriptions and citations to this point.

Response: We thank the reviewer for this excellent suggestion. We have added the following sentences to the collective coupling discussion about mode volume and plasmonic structures on page 15, lines 402-407.

“It is also noteworthy that collective strong coupling can be omitted by using plasmonic structures with radically-reduced mode volumes and therefore increased cavity–vibration coupling strength.[64, 65] Importantly, different from many other polaritonic applications, the presence of small Q values (due to ohmic losses) in such plasmonic implementations may be beneficial rather than detrimental for enhancing the yield of the ESVP mediated down conversion process (cf. Figure 4f).”

(2) It might be kinder to put subscript “c” for p and q used for the cavity (ex. pc and qc as seen in the Supporting Information, page 9 line 4) since authors use them both for cavity and polariton in the Supporting Information.

Response: We agree and have changed the text accordingly.

(3) In Figure S1b, what is the origin of the region $\langle N_{\text{cav}} \rangle > 2.00$ shown in yellow?

Response: We thank the reviewer for raising this question. The region indicated with “ > 2.00 ” is due to coupling of low-energy photons to the vibration. For the vibration, the quantum is fixed at 1600 cm^{-1} whereas the reorganization energy is fixed at 1200 cm^{-1} . The lower the photon energy, the more photonic quanta can be produced, in principle, out of this energy. However, the production of quanta is inhibited by the cavity mode being off-resonance from the vibration. In turn, this is overcome by sufficiently strong cavity-vibrational coupling, as a result of which the cavity becomes highly populated, as observed in Figure S2B. We have expanded the discussion of this figure on page S14-S15, and added another figure (Figure S3) further detailing this.

Reviewer 3’s comments:

The article by Weatherly et al. proposes a hypothetical photonic down conversion process where UV/vis photons are converted into IR radiation via polariton modes generated from the Franck-Condon electronic excitation process of a molecule in an infrared microcavity in the regime of vibrational strong light-matter coupling.

This a timely topic and the proposed idea is new as far as I am aware. However, there are a few points of the theoretical treatment that are unclear, and almost no attention has been paid to some key features of polaritonic systems that would likely make the feasibility of the process hypothesized by the authors to be essentially negligible. If the authors can address these points satisfactorily, then I would probably recommend publication at this journal. The specific issues that should be addressed are:

Response: We thank the referee for appreciating the timeliness of this topic and the novelty of our proposed idea. We furthermore thank the reviewer for the detailed comments and suggestions, in particular those addressing the feasibility of the proposed process, which we share our perspective on in the following. At the same time, we should emphasize that this work breaks new ground in excited-state vibrational polaritons

(ESVPs) broadly speaking, and that as such it delivers important insights for the nascent fields of polariton photochemistry and related areas, extending beyond the downconversion process discussed in our manuscript.

1. The authors have emphasized in their discussion that experiments in the vibrational strong coupling regime occur with a large ensemble of molecules, yet their modeling always considers only a single molecule with unrealistically large values of single-molecule coupling strength. There is no argument put by the authors on whether the efficiencies they compute in Figs 3 and especially 4 are expected to persist when an ensemble of molecules interact with the cavity mode and a large density of dark states result from that. In fact, simple arguments in the literature suggest that any time a large density of molecules is considered effects obtained from single-molecule models such as those reported by the authors become irrelevant. If there is a reason to expect that the downconversion suggested by the authors will not have infinitesimal efficiency under ensemble conditions, the authors should clarify that, and, if there is no reason the proposed effect would ever be even observable, then, this publication is probably better suited for a more specialized journal.

To be clear, based on current literature (e.g., Feist, Subotnik, Yuen-Zhou), the impression one has is that within a single-photon mode description of the cavity, a generalization of the current approach to include many molecules and the resulting dark states would lead to an extremely small efficiency of the downconversion process because the electronic excitation from the ground-state would have much larger probability to excite dark states than polaritons and because polariton energy dissipation would also direct significant amounts of more energy to dark states reducing even more the fraction of IR energy coming from the microcavity. If this argument doesn't apply here, then it would be important to get the authors perspective on these points since these are ideas currently permeating the polariton literature.

Response: We thank the reviewer for raising this comment. We also acknowledge that there remains a general lack of consensus on how exactly any of the strong coupling phenomena discussed in the literature, including ground state polariton chemistry, play out against a reservoir of dark states. We furthermore acknowledge that under a single-photon mode such a reservoir seems to generally inhibit polaritonic behaviors. This of course leaves the all-encompassing question of how to rationalize the high efficacies of polaritonic phenomena observed in experiments. Our work is not intended to answer this question. Instead, in its aim to break new ground in ESVPs it follows an approach similar to the majority of theoretical works in other polaritonic subfields that restricted themselves to the single-molecule limit, which have nonetheless captured many of the experimentally-observed trends, and which have provided much of the basis for our understanding of polariton chemistry. The wealth of experimental evidence that ground state chemical reactions can be impacted by vibrational polaritons gives a strong precedence for similar effects in the excited states by means of ESVPs. Through our approach we aim to contribute valuable mechanistic insights into ESVPs that by themselves are immune to the possible effects of collective strong coupling, such as the importance of maximizing the Huang-Rhys factor and excited state IR dipole, the result of a cavity emission maximum at a specific quality factor, and the requirement that generating ESVP by photo-excitation requires molecules lacking inversion symmetry. These insights lay the theoretical basis needed for exploring additional complexities such as collective coupling effects and disorder. As such, we believe to present our findings in a way that is particularly useful to a broader audience, and which hopefully stimulates further experimental and theoretical research into ESVPs.

Having said that, we would also like to share a number of considerations that makes us optimistic about the feasibility of the ESVP-mediated downconversion process proposed in our work. While detrimental effects of dark states have previously been demonstrated for models based on single-particle polaritonic Hamiltonians involving large numbers of molecules, much remains to be learned about the case involving multi-particle polaritonic states, and it is conceivable that increasing the ratio of particles to molecules offsets these effects. The down conversion process proposed here can manifest as a multi-particle phenomenon even for a single molecular excitation, provided that the Huang-Rhys factor is sufficiently large. It is furthermore possible to increase the particles-to-molecules ratio by the decoupling of ground state vibrations (through state-selective IR dipoles, or by detuning ground state modes from the cavity through state-selective vibrational frequencies), or by sheer depletion of ground state molecules by using strongly absorbing molecules and high laser fluences. It should also be mentioned that a completely different avenue for the implementation of the down-

conversion process is provided by the use of plasmonic structures with small mode volumes, as also pointed out by Reviewer 2, which allow for only a small number of molecules to couple to the cavity but which also achieve the required coupling strength to realize the downconversion process.

In response to the reviewer’s comment we have expanded the discussion of our results on pages 15, lines 382-407.

2. The authors use the Hamiltonian including the counterrotating-wave terms but ignore the A^2 term on the basis that the latter should only be relevant under ultrastrong coupling. However, the same argument used to ignore A^2 terms is often used to ignore the counterrotating as it happens frequently in models of strong light-matter coupling. Thus, it seems inconsistent to ignore one type of term and not the other. The authors should clarify this. What is the effect of the A^2 term? If the computational cost is not larger, then why not do the calculation with the full Hamiltonian? In fact, I think the full Hamiltonian calculations should be given at least in the Supplementary Material, because if they turn out to lead to significantly different results than the computations without the A^2 term, that could point to an issue that would have to be addressed.

Response: We originally excluded the A^2 term from our calculations in order to simplify the discussion of our results. The counter-rotating terms, on the other hand are trivially accounted for within the TWA, and were therefore included. However, we appreciate the inconsistency of ignoring the A^2 term while including the counterrotating terms, and in our re-submission we have included a comparison with and without the A^2 term, showing the negligible differences for the applied coupling values. This is presented in section 4 in the SI, and a reference has been added from the main text on page 5, lines 145-146.

3. At OK the authors mention that regardless of the value of the light-matter coupling, the initial photon population in the cavity is zero. Given that counterrotating terms are included in the Hamiltonian, wouldn’t the cavity-matter composite ground-state have a non-zero photon occupation number that also increases with the light-matter coupling? Did the authors ignore this non-zero photon occupation number in the ground-state when calculating $\langle N_{\text{cav}}(t) \rangle$? If yes, then needs to be explained, and if no, then the reason there is zero photonic content in the total ground-state should also be discussed since this does not seem like a generic feature of models including counterrotating terms.

Response: We thank the reviewer for raising this question, which highlights a source of confusion in our initial submission. To clarify, we did include the non-zero photon occupation number in all of our results and discussed this effect in lines 180-194 of the initial submission (lines 192-197 in the revised manuscript):

“For weak coupling, where $\Omega_\gamma \approx \omega_c = \omega_v$, it is apparent that $N_\beta \approx 0$. But, larger coupling strengths result in $N_\beta > 0$, which is concurrent with the breakdown of the RWA. In that case, the polaritonic ground state within the RWA, which has no photons or vibrational quanta, mixes with the state containing a single cavity photon and single vibration, and therefore, even at zero temperature there will be a non-zero number of cavity photons if the coupling is large.”

However, at line 203 we previously stated that

“At time zero, $\langle \hat{N}_{\text{cav}} \rangle = 0$ for all values of g since the UV/vis excitation acts exclusively on the electronic DOF, inducing vibrations on the excited state.”,

which is only approximately true as there is some very small number of photons in the ground state due to the counter-rotating terms. We have re-written the sentence as (page 9, lines 206-209)

“At time zero, $\langle \hat{N}_{\text{cav}} \rangle \approx 0$ for all values of g shown since the UV/vis excitation acts exclusively on the electronic DOF, inducing vibrations on the excited state. Note that there is a small but non-zero number of cavity photons at time zero due to the effect of counter-rotating terms captured in N_β .”.

Other changes made to the Supplemental Information

We have also made minor corrections to equations S23, S24, S30, S31, S32 in the SI. Previously in these equations we had forgotten to add a prefactor (the squared ratio of the vibrational frequency to the polariton frequency) to the last term in each equation. These changes do not effect any of the results presented in the manuscript or SI.

REVIEWERS' COMMENTS

Reviewer #2 (Remarks to the Author):

The authors have addressed most of the concerns raised by the reviewers, however, the issue regarding the effect of large molecular ensembles raised by reviewers 1 and 3 and is an important one. The authors confine attention to a single chromophore but the only way to obtain sufficient radiation-matter coupling (in an IR cavity) is to have large numbers of molecules. The author's response includes the possibility of maintaining a single chromophore but using reduced mode volumes (i.e. using plasmonic cavities) to increase coupling strength. This is indeed possible, but there may be additional complications since the cavity needs to be tuned to an IR mode. This being said, I believe the novelty of the effect - even if it is confined to a single molecule- justifies publication and will likely motivate additional works to investigate the effects of larger molecular ensembles.

Reviewer #3 (Remarks to the Author):

The manuscript has been revised satisfactorily. For the discussion of plasmonic structures, I would suggest the authors add citations such as a review by P. Törmä and W. L. Barnes (Rep. Prog. Phys.,2015, 78, 013901), papers on vibrational strong coupling with surface plasmon by the Barnes group (ACS Photonics 2019, 6, 8, 2110–2116) and by the Shegai group and the Börjesson group (Nano Lett. 2021, 21, 3, 1320–1326), and a recent arxiv article by the Barnes group and the Baumberg group (arXiv:2304.04834 [physics.optics])

Reviewer #4 (Remarks to the Author):

I recommend publication as the authors have addressed satisfactorily all posed questions.

Reviewer #2 (Remarks to the Author):

The authors have addressed most of the concerns raised by the reviewers, however, the issue regarding the effect of large molecular ensembles raised by reviewers 1 and 3 and is an important one. The authors confine attention to a single chromophore but the only way to obtain sufficient radiation-matter coupling (in an IR cavity) is to have large numbers of molecules. The author's response includes the possibility of maintaining a single chromophore but using reduced mode volumes (i.e. using plasmonic cavities) to increase coupling strength. This is indeed possible, but there may be additional complications since the cavity needs to be tuned to an IR mode. This being said, I believe the novelty of the effect - even if it is confined to a single molecule- justifies publication and will likely motivate additional works to investigate the effects of larger molecular ensembles.

We appreciate the reviewer's diligent effort in reviewing our work and for providing further feedback. We agree with the reviewer on the importance of studying this process in collectively coupled systems and hope our work can act as the foundation for future investigation in this direction. The reviewer further suggests there may be additional complications arising when tuning plasmonic structures to the IR region. As far as we are aware, these complications are practical rather than fundamental. In that regard, it is reassuring to learn about the references provided by reviewer 3, showing that IR active plasmonic structures have been realized and that their coupling to molecular vibrations has been demonstrated to decrease the number of emitters required to reach strong coupling. Accordingly, we have added the following sentence:

“Although there may be practical complications arising when tuning plasmonic structures to the IR spectral region, previous demonstrations of vibrational-strong coupling with IR active plasmonic structures support the feasibility of this strategy.”,

and included reviewer 3's suggested references (see revised manuscript, page 15, lines 422-423).

Reviewer #3 (Remarks to the Author):

The manuscript has been revised satisfactorily. For the discussion of plasmonic structures, I would suggest the authors add citations such as a review by P. Törmä and W. L. Barnes (Rep. Prog. Phys.,2015, 78, 013901), papers on vibrational strong coupling with surface plasmon by the Barnes group (ACS Photonics 2019, 6, 8, 2110–2116) and by the Shegai group and the Börjesson group (Nano Lett. 2021, 21, 3, 1320–1326), and a recent arxiv article by the Barnes group and the Baumberg group (arXiv:2304.04834 [physics.optics])

We thank the reviewer for sharing their insights, especially regarding plasmonic structures and their potential applicability towards this work. We have added the suggested references along with the following sentence (see revised manuscript, page 15, lines 422-423).:

“Although there may be practical complications arising when tuning plasmonic structures to the IR spectral region, previous demonstrations of vibrational-strong coupling with IR active plasmonic structures support the feasibility of this strategy.”

Reviewer #4 (Remarks to the Author):

I recommend publication as the authors have addressed satisfactorily all posed questions.

We thank the reviewer for their time, effort and insights on this review.

Other changes made to the manuscript:

We have made a minor modification to Figure 2 to improve clarity. We have added dashed lines that represent the polariton potential energy surface as well as lines on both polariton axes to indicate the potential minimum along these axes. We believe these changes to give a more intuitive picture of the system.

We have made a minor correction to equations 15, 27, 28, and 29 in order to improve clarity. The total Hamiltonian given in equation 14 uses the bilinear coupling constants κ_i and χ_i which are used when the coupling is described by the product of position operators of the two coupled systems. In the methods section we relate these coupling constants to the quality factor; however, the relationship is for coupling constants used when the coupling is described by raising and lowering operators of the coupled systems after the rotating wave approximation has been applied (Jaynes-Cummings like coupling constants). We now differentiate the Jaynes-Cummings-like coupling constants with a prime, as now given in equations 27, 28, and 29, and equation 15 now includes the prefactor that relates the bilinear coupling to the Jaynes-Cummings constant. These changes do not affect any results presented in the manuscript or SI.